# Cultivation of *Saccharomyces cerevisiae* with Feedback Regulation of Glucose Concentration Controlled by Optical Fiber Glucose Sensor

**DOI:** 10.3390/s21020565

**Published:** 2021-01-14

**Authors:** Lucie Koštejnová, Jakub Ondráček, Petra Majerová, Martin Koštejn, Gabriela Kuncová, Josef Trögl

**Affiliations:** 1Institute of Chemical Process Fundamentals of the CAS, v. v. i., Rozvojová 135/1, 16502 Prague, Czech Republic; ondracek@icpf.cas.cz (J.O.); majerova@icpf.cas.cz (P.M.); kostejn@icpf.cas.cz (M.K.); kuncova@icpf.cas.cz (G.K.); 2Faculty of Environment, Jan Evangelista Purkyně University in Ústí nad Labem, Pasteurova 3632/15, 40096 Ústí nad Labem, Czech Republic; Josef.Trogl@ujep.cz

**Keywords:** yeast cultivation, feedback regulation, glucose detection, optical biosensor

## Abstract

Glucose belongs among the most important substances in both physiology and industry. Current food and biotechnology praxis emphasizes its on-line continuous monitoring and regulation. These provoke increasing demand for systems, which enable fast detection and regulation of deviations from desired glucose concentration. We demonstrated control of glucose concentration by feedback regulation equipped with in situ optical fiber glucose sensor. The sensitive layer of the sensor comprises oxygen-dependent ruthenium complex and preimmobilized glucose oxidase both entrapped in organic–inorganic polymer ORMOCER^®^. The sensor was placed in the laboratory bioreactor (volume 5 L) to demonstrate both regulations: the control of low levels of glucose concentrations (0.4 and 0.1 mM) and maintenance of the glucose concentration (between 2 and 3.5 mM) during stationary phase of cultivation of *Saccharomyces cerevisiae*. Response times did not exceed 6 min (average 4 min) with average deviation of 4%. Due to these regulation characteristics together with durable and long-lasting (≥2 month) sensitive layer, this feedback regulation system might find applications in various biotechnological processes such as production of low glucose content beverages.

## 1. Introduction

The number of people with diabetes has risen from 108 million in 1980 to 422 million in 2014, and between 2000 and 2016 there was a 5% increase in premature mortality from diabetes the World Health Organization (WHO) predicts that the diabetes will be the seventh leading cause of death in 2030 [1]. The accurate evaluation of the glucose content in foods is extremely important for the maintenance of its physiological level in blood of diabetic individuals. Information about glucose content of foods and beverages is essential for both producers and consumers. Glucose monitoring is crucial in tracing the fermentation processes in the wine, brewing, and dairy industries.

The first qualitative test of glucose was published in 1848 [2]. Since that time many methods of glucose quantification have been described [3]. Web of Science links to 25,000 references for key words glucose detection. Plenty of physical detection principles have been used which are often non-specific to glucose [4]. For example, microwave resonator-based sensors might be advantageous for glucose detection in blood [5,6] or in some industrial applications as their linear range of measured glucose concentrations is from zero to more than ten weight percent [7]. In comparison with these microwave-based sensors, optical biosensors with glucose oxidase exhibit a high specificity to glucose.

The first glucose biosensor was realized in 1962 using glucose oxidase and a Clark electrode [8]. Since that time, hundreds of glucose biosensors have been described with electrochemical or optical transducers [9]. Prevailing research effort was focused on biosensors for monitoring concentration of glucose in blood and other physiological fluids [10,11]. Nevertheless, glucose biosensors for beverages and food industry have also been presented. Ayenimo et al. described a polypyrrole-based bilayer amperometric glucose biosensor integrated with a permselective layer, which was successfully employed for glucose determination in various fruit juices [12]. For glucose and galactose detection in fruit juices and skim milk, the graphite working electrode, on which glucose oxidase and β-galactosidase were coimmobilized by means of covalent bonding, was developed by Portaccio et al. [13]. Amperometric biosensor based on modified screen-printed carbon electrodes for online glucose monitoring during cultivation of *Saccharomyces cerevisiae* in microbioreactor was published by Panjan et al. [14]. Otten at al. described a fluorescence resonance energy transfer (FRET)-based glucose biosensor, which can be applied in microbioreactor-based cultivations. The soluble sensor was successfully applied online to monitor the glucose concentration in an *Escherichia coli* culture [15].

Optical fiber sensors have showed a number of advantages over electrochemical sensors due to their independence of electromagnetic involvement, security, sensitivity, ruggedness, and fine dimensions of probes. In the last twenty years, the progress in the development of optical fiber sensors of glucose was reviewed by Wolfbeis [16,17,18,19,20] and Wang [21,22,23]. Among various principles of optical fiber sensing of glucose, enzymatic sensors with glucose oxidase and optical oxygen transducers have been the most broadly studied and used. The combination of high selectivity of glucose oxidase and ruggedness of optical oxygen transducer allow them to be applied in industrial processes.

Except for fast and precise detection, food and biotechnology processes require low deviations from desired glucose concentration (*c_GL_^DES^*) and their quick compensation. An advantageous solution to these demands is an incorporation of biosensoric detection components into feedback loops, which keep actual glucose concentration (*c_GL_*) on desired value (*c_GL_* = *c_GL_^DES^*), or in allowed limits throughout production process. A continuous measurement of glucose in real-time without connection to glucose dispenser was described by Maldonado et al. [24] and Blankenstein et al. [25].

The system of control of glucose concentrations of a perfusion medium in a rotating wall perfused vessel bioreactor culturing BHK-21 cells was presented by Xu et al. in 2004. The custom-made glucose sensor was based on a hydrogen peroxide electrode. The system first controlled the glucose concentration in perfusing medium between 4.2 and 5.6 mM for 36 days and then at different glucose levels for 19 days. A stock solution with a high glucose concentration (266 mM) was used as the glucose injection solution. The standard error of prediction for glucose measurement by the sensor, compared to measurement by the Beckman glucose analyzer, was 0.4 mM for 55 days [26]. Commercially available systems of glucose control in bioreactors (CITSens Bio, SEG-Flow) use the TRACE filtration probe for harvesting cell-free filtrate from bioreactors and fermenters under sterile conditions. Company Stratophase Ltd. (Hampshire, United Kingdom) developed a sensor that regulated glucose concentration with an in situ optical glucose sensor. The sensor with Bragg grating measures glucose concentrations as changes of refractive index. Such measurements are nonspecific and, therefore, they suffer from great errors due to changes of concentrations of the other medium components.

Enzymatic glucose sensor with oxygen transducer, its preparation and analytical features, were described in previous papers [27,28,29]. The sensor withstands sterilization by UV and ethanol as well as mechanical stresses caused by mixing of fermentation broth. The sensor is based on the measurement of oxygen consumption due to oxidation of glucose catalyzed by an enzyme, glucose oxidase. Ruthenium complex serves as an optical transducer. Its fluorescence is quenched proportionally with oxygen concentration. Both sensitive parts—the enzyme (preimmobilized on Sepabeads^®^) and the complex—are coated in Ormocer^®^ (organic–inorganic polymer). The polymer has a siloxane network and is UV curable. Covalent attachment to a carrier (Sepabeads^®^) protects the enzyme against harsh conditions after mixing with Ormocer^®^ and during the UV curing of the sensitive layer on the acrylate lens.

Here, we present on-line feedback regulation of glucose concentration controlled by this glucose sensor placed in the bioreactor vessel. The regulation was demonstrated in both modes: in dilution and fed batch cultivation. Dilution mode was proposed for production of beverages with limited content of glucose and for diabetics, where glucose content should be close to zero. In fed-batch cultivation, microorganisms consume glucose, but to keep cells alive and producing, the concentration of glucose must be maintained at a level which allows cells to survive but limits their proliferation. In the fed batch process, glucose regulation is demonstrated with the most industrially used microorganism, *Saccharomyces cerevisiae.*

## 2. Materials and Methods

### 2.1. Chemicals

D-glucose p.a., sucrose p.a, glutaraldehyde 25%, K_2_HPO_4_⋯3 H_2_O p.a., and KH_2_PO_4_ p.a. were purchased from Penta s.r.o. (Czechia). Yeast extract, glucose oxidase type X-S from *Aspergilus niger* with specific activity 228.4 kU/g (GOX^X-S^), and glucose oxidase type II-S from *Aspergilus niger* with specific activity 37.7 kU/g (GOX^II-S^) were from Sigma Aldrich^®^ (St. Louis, MO, USA). Bacteriological pepton was purchased from Oxoid-Thermo Fisher Scientific Inc. (Waltham, MA, USA). NaOH p.a. and H_3_PO_4_ 85% were purchased from Lach-Ner s.r.o. (Prague, Czechia). Sepabeads^®^ EC-HA 403 (SEPA) were delivered by Resindion S.r.l (Binasco, Italy). Tris (4,7-difenyl-1,10-fenantrolin) ruthenium(II) dichloride (RuC) was purchased from ABCR GmbH (Karlsruhe, Germany). Ormocer^®^ KSK 1248 (ORM) was obtained from Fraunhofer Institute for Silicate Research ISC (Wurzburg, Germany). Photoinitiator Irgacure^®^ 500 was from BASF, Germany. Polymethylmethacrylate biconvex lens with diameter 7 mm was from Institute of Plasma Physics of the CAS, v.v.i. (Prague, Czechia).

### 2.2. Media and Gasses

Potassium phosphate buffer (50 mM, *pH* 7) was prepared by dilution K_2_HPO_4_⋯3 H_2_O (21.1 mL) and KH_2_PO_4_, (28.9 mL) in distilled water (1 L). Final *pH* 7 was adjusted by addition of NaOH (0.2 M) or H_3_PO_4_ (0.2 M).

YPG cultivation medium contained yeast extract (1 g), peptone (2 g) and glucose (2 g) in distilled water (100 mL).

Concentrated glucose solution (1 mM) was prepared by dilution glucose (90 g) in distilled water (500 mL).

Oxygen, 2.5 UN1072 and nitrogen 4.0, UN1066 were products by Linde Gas a.s. (Prague, Czechia).

### 2.3. Microorganisms

*Saccharomyces cerevisiae* was obtained from Collection of microorganisms of the Institute of Biochemistry and Microbiology UCT Prague. Overnight culture (50 mL) was added into the bioreactor. Optical density (*OD*) of the overnight culture (3× diluted) was 0.5.

*OD* was determined in 1 cm cuvette at 600 nm by UV–VIS spectrophotometer HP8452A (Hewlett-Packard, Palo Alto, CA, USA).

### 2.4. Preparation of Optical Sensitive Layers

Sensitive layers were prepared by procedure described in details by Kostejnova et al. [29]. Briefly, 200 mg of SEPA was activated by a stirring with glutaraldehyde (4 mL) for four hours. After centrifugation and washing, the enzyme solution was added to activated SEPA and the mixture was stirred for 18 h with the same velocity as was used during activation. The mixture was centrifuged, the supernatant was removed, and Sepabeads^®^ with immobilized glucose oxidase (SEPA-GOX) were washed twice with buffer. Ormocer^®^ was mixed with Ru complex, Irgacure 500 and sucrose to form ORM-RC. Components of sensitive layers were mixed on glass slide in the ratio 2:1, ORM-RC: SEPA-GOX. The mixtures were deposited on plastic lenses and cured by UV light for ten minutes. After UV polymerization the lenses were immersed in phosphate buffer (50 mM, *pH* 5.9) overnight to wash out sucrose. The thicknesses were measured with the microscope Tescan (Czech Republic) in the center and on the periphery of the layers (see Figure 1). Parameters of preparation and analytical characteristics (sensitivity (*SN*), linear dynamic range (*LDR*), and response time (*RT*) of sensitive layers used in the tests are presented in Table 1 and on Figure 1.

### 2.5. Feedback Regulation System

A schema with photos of the feedback regulation system is on Figure 2.

The bioreactor (Figure 2, position 2) was produced by Applicon V.B. (Schiedam, The Netherland), it is autoclavable, comprised of a tempered glass vessel with volume of 5 L and inner diameter 166 mm equipped with Bio controller ADI 1010 and Bio console ADI 1025 for control of *pH* (pH electrode), the concentration of dissolved oxygen (*dO2*) is measured with Clark electrode, has temperature control (resistance thermometer), and monitors the velocity of mixing.

Optical probe for measurement of glucose concentration had identical shape as *pH* and *dO2* probes of the bioreactor (Figure 2, position 3–4). The probe consisted from stainless steel tube with glucose sensitive layer, and a bundle of optical fibres connected to the light source and the detector (Figure 2, position 5). NeoFox Sport made by Ocean Optic (Largo, FL, USA) was used as light source and the detector. Increase of glucose concentrations in the reactor was detected with sensitive layer on acrylate lens (Figure 2. position 4) as the increase of fluorescence lifetime of RuC due to consumption of oxygen in oxidation of glucose catalyzed by glucose oxidase.

Analog output signal from NeoFox was read by serial port of the control unit (Figure 2. position 8). After evaluation with control software (Figure 2, position 7) the output signal was controlling rotation speed of peristaltic pump Masterflex 7518-00 (Cole-Parmer Instrument Company, Vernon Hills, IL, USA) (Figure 2, position 9). Measurement and control software was developed in Labview software environment. The Labview control software allows for following settings: (1) desired glucose concentration *c_GL_^DES^*, which was set as life time set point; *τ^DES^* (2) a mode of concentration control—diluting or feeding; (3) time resolution of glucose concentration measurement—averaging time, *t_checII_*; and (4) speed of glucose dosing with peristaltic pump as percentage of pump power. Actual measured glucose concentration and indication of pump action (run/stop) were displayed on the monitor. The software recorded measured glucose concentrations *c_GL_* and course of administration of buffer or concentrated solution of glucose. The system responded on deviation from *c_GL_^DES^* after checking time (*t_chec_*). During *t_chec_* the system only measures without responding. In dilution mode, the pump administrated buffer in case that *c_GL_* was the same or higher than maximum allowed glucose concentration (*c_GL_^MAX^*). In cultivation mode, the concentrated glucose solution was added into the bioreactor in case that *c_GL_* was the same or lower than minimum allowed glucose concentration (*c_GL_^MIN^*). To reach desired concentration, *c_GL_* = *c_GL_^DES^*, the volume of a dose (buffer/conc. glucose solution) was calculated by the software with respect of volume of liquid in the bioreactor. The new *t_chec_* started after the pump finished dosing.

### 2.6. Reproducibility of Biosensor Response in Repetitive Measurements during 2 Months

The probe of the biosensor with sensitive layer (enzyme concentration 125 mg GOX^X-S^/g SEPA, thickness of the layer 300 nm) was immersed in non-sterile buffer (50 mM, *pH* 7), which was bubbled by sterile air with volume flow 16 mL/min, mixed 400 rpm, and tempered 25 °C. During two months, on working days, *SN*, *LDR*, and *RT*. were determined once a day. Measurements and calculations of analytical characteristics are described in details in previous paper [29].

### 2.7. Sterilization

The bioreactor filled with medium/buffer with inserted *pH*, *dO2*, *T* probe, together with storage bottles of base, acid, concentrated glucose, dilution buffer, and all connection pipes was sterilized in autoclave at 120 °C for 30 min. Before inserting into bioreactor, glucose probe, and acrylate lens with sensitive layer were sterilized by immersing in ethanol (70%) for 5 min and irradiation with UV for 10 min. 

### 2.8. Off-Line Measurement of Glucose Concentration

Off-line glucose concentration was measured with Glucose oxidase Activity Assay kit from Sigma Aldrich s.r.o. (Prague, Czechia).

### 2.9. Control of Glucose Concentration with Feedback Regulation System

#### 2.9.1. Feedback Regulation of Glucose Concentration to Lower Level (Dilution Mode).

The bioreactor filled by 2 L buffer (50 mM, *pH* 7) was bubbled by sterile air with volume flow 16 mL/min, mixed 400 rpm, and tempered 25 °C. The value of *dO2* in the bioreactor was 21%. The regulation was tested at two maximum concentrations, *c_GL_^MAX^ =* 0.5 mM and 0.125 mM for corresponding *c_GL_^DES^* = 0.4 mM and *c_GL_^DES^* = 0.1 mM, respectively.

NeoFox was switch on and the system was left to stabilize fluorescence lifetime (*τ^0^*) at zero glucose concentration (*c_GL_^0^*) for 15 min. To calibrate Neofox, glucose concentration was increased by addition of concentrated glucose solution to *c_GL_^DES^* = 0.4 mM or 0.1 mM (Figure 3 and Figure 4. green frames), which lead to increase of fluorescence lifetime (*τ^DES^*). On user interface monitor were set *τ^0^, τ^DES^* and corresponding *c_GL_^0^*, *c_GL_^DES^*, times for averaging *t_chec_* = 10 min and *c_GL_^MAX^ =* 0.5 mM resp. 0.125 mM (Figure 3 and Figure 4, red frames) and corresponding *τ^MAX^* calculated from the calibration. After three *t_chec_* (Figure 3 and Figure 4, position 1) concentration of glucose was increased from *c_GL_^DES^* = 0.4 mM resp. 0.1 mM to *c_GL_^MAX^ =* 0.5 mM resp. 0.125 mM by hand pipetting of solution of concentrated glucose (0.2 mL, resp. 0.05 mL) into the bioreactor (Figure 3 and Figure 4, position 2). After *t_chec_*, *c_GL_^MAX^* was detected and the pump of feedback loop dosed calculated buffer volume into the bioreactor to reach *c_GL_* = *c_GL_^DES^* (Figure 3 and Figure 4, position 3). In reality, glucose concentration after regulation (*c_GL_^REG^*) differs from *c_GL_^DES^.* The cycles of addition of glucose solution and regulation were repeated three times during both tests. The test for *c_GL_^DES^* = 0.4 mM was reproduced three times (Figure 3.I–III).

Response time (*RT_90_*) was calculated for each buffer dose according to equation
*RT_90_* = *t_1_* − *t_2_*,(1)
where *t_1_* is time when actual measured glucose concentration exceeded glucose concentration after regulation for 10% *c_GL_* = *c_GLREG_ +* 0.1 × *c_GLREG_*, and *t_2_* is time when buffer was added.

Deviation (*s*) of glucose concentration after regulation from *c_GL_^DES^* was calculated for each buffer dose
*s* = ((*c_GL_^DES^* − *c_GL_^REG^*)/*c_GL_^DES^*) × 100.(2)

#### 2.9.2. Feedback Regulation of Glucose Concentration of Fed Batch Cultivation of *Saccharomyces cerevisiae* in Stationary Phase (Cultivation Mode)

The bioreactor was filled with 2 L incomplete YPG medium. Throughout the experiment, the bioreactor was tempered to 30 °C and bubbled with sterile air, oxygen, or nitrogen to keep constant *dO2* = 21%. Concentration of glucose in incomplete YPG medium was measured off-line. Double point glucose calibration was done in the first hour of the experiment. The first point was the concentration of glucose in incomplete YPG medium *c_GL_^MIN^* (2 mM, Figure 4, red frame) and the second point was *c_GL_^DES^* = 3.5 mM (Figure 4, green frame) acquired by hand pipetting of concentrated glucose (3 mL). Neofox corresponding fluorescence lifetimes, *τ^MIN^* and *τ^DES^*, were set on user software monitor together with *t_chec_*. Double the response time (*RT_90_*), determined in calibration, was opted for checking time, *t_chec_* = 10 min.

After calibration, glucose (40 g) was added to complete YPG medium, thus *c_GL_ =* 111 mM, which was out of the range (0–7 mM) of the biosensor (Figure 5, position 1). The bioreactor was inoculated with night culture of *Saccharomyces cerevisiae* (50 mL, *OD* for 3x diluted culture was 0.5) and feedback regulation was switched on (Figure 5, position 2). The growing cells consumed glucose. After 11.5 h of fermentation, *c_GL_* dropped below 7 mM (Figure 5, position 3). Within 6 *t_chec_*, the measured glucose concentration *c_GL_* decreased from 6.8 to 1.9 mM and *c_GL_* < *c_GL_^MIN^* (Figure 6. position 1). At the end of 6th *t_chec_* the pump started to dose concentrated glucose solution into the reactor so that *c_GL_* = *c_GL_^DES^, resp. c_GL_^REG^,* after the seventh *t_chec_* (Figure 6, position 2). Culture of *Saccharomyces cerevisiae* consumed added glucose during the 8th *t_chec_* and *c_GL_^DES^*, and *c_GL_^REG^* dropped to (or under) *c_GL_^MIN^*, which activated dosing pump (Figure 6, position 3). The cycle kept adding glucose to reach *c_GL_^DES^* followed by consumption with yeast culture to *c_GL_^MIN^* (cycle ↓↑) was repeated seven times. The experiment was twice reproduced.

Response times were calculated for each glucose dose according to
*RT_90_^*^* = *t_1_* − *t_2_*, (3)
where *t_1_* is time when measured glucose concentration reach 90% of concentration after regulation: *c_GL_* = 0.9. *c_GL_^REG^* and *t_2_* is time when concentrated glucose solution was added.

Deviation (*s^*^*) from *c_GL_^DES^* were calculated for each glucose dose according to
*s^*^* = (*c_GL_^DES^* − *c_GL_^REG^*)/*c_GL_^DES^* × 100.(4)

## 3. Results and Discussion

### 3.1. Feedback Regulation of Glucose Concentration to Lower Level (Dilution Mode)

This regulation is demonstration of application of feedback system in production of beverages for diabetes, where the demand is to keep glucose concentration at a level close to zero. The sensitive layer, used in dilution mode, was chosen to meet the need of the lowest detection limit. Based on our previous study [29], such demand best fit sensitive layer comprising high content of enzyme, which is immobilized on undivided SEPA. 

Monitoring and control of low glucose concentration levels are in Figure 3 and Figure 4, and characteristics of regulation are in Table 2. In all experiments, response times were shorter than 5 min (*RT_90_* ≤ 5 min). Average deviation was 1.8% for glucose concentration hold at 0.4 mM. For lower glucose concentration, the relative precision of measurement decreased. Therefore, in case the desired glucose concentration was equal to limit of detection (*LOD*) of used biosensor (*c_GL_^DES^* = *LOD* = 0.1 mM, the test IV.), the average deviation increased to 3.7%.

### 3.2. Feedback Regulation of Concentration of Glucose of Fed Batch Cultivation of Saccharomyces Cerevisiae in Stationary Phase (Cultivation Mode)

In cultivation mode, the sensitive layer should possess fast response time and wide concentration range to measure glucose in sufficiently broad concentration range during stationary phase of cultivation. In our previous paper [29], it was shown that *LDR* of the layers increased with decreasing enzyme concentration. It was also shown that crushing of spherical SEPA with immobilized glucose oxidase resulted in higher *LDR*. Unfortunately, increasing *LDR* simultaneously increased *RT*, which is an undesirable effect for the feedback regulation system. Therefore, we must compromise between opposing demands on analytical features of sensitive layer for cultivation mode. We used the sensitive layer with *RT* ≤ 6 min and *LDR* = 0–7 mM.

A time record of complete cultivation of *Saccharomyces cerevisiae* is presented on Figure 5 and the detail of stationary phase, while glucose concentration was controlled with the feedback regulation system, is on Figure 6. Table 3 shows that in all seven cycles ↓↑, response times were below 6 min (*RT_90_^*^* ≤ 6 min) and deviation from regulation did not exceed 9%. The average *RT_90_^*^* was 4 min and the average deviation 3.9%.

In situ monitoring and control glucose concentration during cultivation were described by Tric et al. [30]. They used also enzymatic sensor with optical glucose transducer; however, glucose oxidase was fixed on optically isolated oxygen sensor with glutaraldehyde and covered by perflorated hydrophilic membrane. In comparison with this report, where response times were 6 min for increasing and 10 min for decreasing of glucose concentrations, we reached response times shorter than 6 min in all tests. The shorter response times might be related to faster diffusion of oxygen and glucose in sensitive layer comprising both enzyme and fluorescent complex in one mixture. These results implicate that regulation response times less than few minutes are hard to reach with enzymatic glucose sensor with optical oxygen transducer. Response times become shorter as the activity of enzyme increases and sensitive layer is thinner [26]. Nevertheless, these parameters are limited by technical feasibility of a preparing such layer. Selectivity, robustness, and long-term reliability are favored features of enzymatic glucose sensors with oxygen transducers for control of glucose concentrations in biotechnological processes but, if one minute or less response times are necessary, another type of glucose sensor should be used.

Activity of microorganisms resulted in *c_GL_* decreased from 6.8 to 1.9 mM (*c_GL_* < *c_GLMIN_*, position 1). After this point, the pump started to dose concentrated glucose solution into the reactor so that *c_GL_* = *c_GL_^DES^* (position 2). Culture of *Saccharomyces cerevisiae* continued to consume glucose and *c_GL_* dropped to *c_GLMIN_*, which activated dosing pump again (position 3). *t_1_^*^* is time when measured glucose concentration reached 90% of concentration after regulation and *t_2_^*^* is time when concentrated glucose solution was added.

### 3.3. Reproducibility of the Biosensor Response during 2 Month.

During two months (42 measurements) the average *SN* was 0.306 μs L mmol^−1^ with relative deviation 10% (Figure 7) and an average maximum of linear dynamic range (*LDR^MAX^*) 1.6 mM with relative deviation 12% (Figure 8). At the first measurement, *RT* was 9 min. In the second measurement, *RT* increased to 14.7 min and this response time was preserved in following 40 measurements. An average *RT* (without the first day) was 15.1 min with relative determinative deviation 8% (Figure 9) and it remained constant throughout the repetitions (*p* > 0.9971). After the first experiment, an increase of *RT* is probably a result of an adsorption of microorganisms from non-sterile buffer, which cause diffusion slowdown of both substrates glucose and oxygen, in the sensitive layer.

### 3.4. Wider Applicability of the Biosensor

The presented biosensor was developed with immobilized glucose oxidase aiming for the on-line monitoring of glucose concentration. Together with oxygen and pH, glucose concentration is one of the most often measured parameters in biotechnology. Nevertheless, the presented concept is general and replacing of glucose oxidase by other oxidases can result in various analogical biosensors, such as for biological amines [26] or cholesterol oxidase [27] for use on continuous systems. Of interest in near future might be sensors of various environmental pollutants. Biodegradation pathways of many organic pollutants often start with oxygenases enzymes [28,29] of different specificity, and these could serve as a biosensing elements for regulation of continuous water treatment processes.

## 4. Conclusions

In this work, we presented the feedback system for regulation of glucose concentration based on the enzymatic sensor with optical oxygen transducer. The system was demonstrated for the case of maintaining low glucose concentration. An undesirable increase of glucose concentration was compensated below 0.125 mM by dilution in less than 5 min. In stationary phase of fed batch cultivation when glucose was continuously consumed by growing microorganisms, the feedback system adjusted glucose concentration to 3.5 mM in less than 6 min after detection of the concentration drop to 2 mM. The two-month stability and reproducibility of biosensor response was demonstrated by daily measurements, in which relative determinative deviations of analytical characteristics (sensitivities, linear dynamic ranges, and response times) were less than 12%. In comparison to known and commercially available glucose concentration regulations, the presented feedback system has advantage in use of in situ sensor, robust construction, and long-term stability.

## Figures and Tables

**Figure 1 sensors-21-00565-f001:**
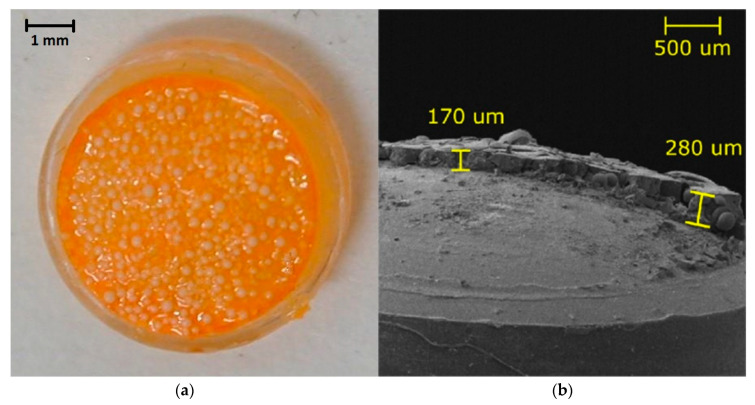
Photos of optical sensitive layer. (**a**) The lens with sensitive layer (used in the cultivation). (**b**) Scanning electron microscopy photo of cross-section of this sensitive layer.

**Figure 2 sensors-21-00565-f002:**
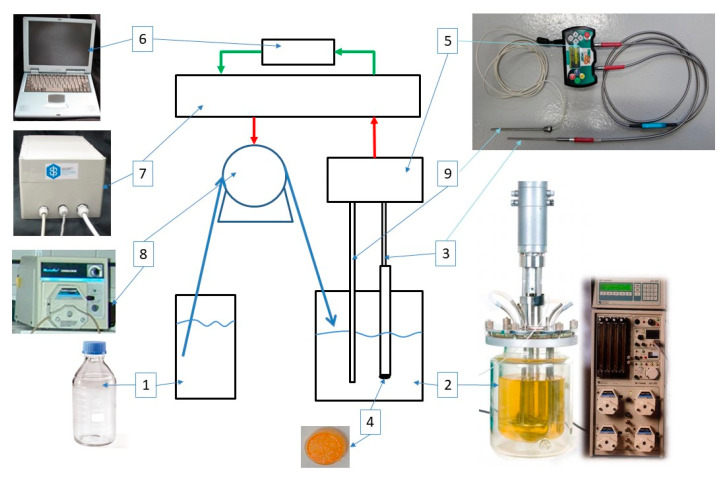
Feedback regulation system. 1. container with buffer or concentrated glucose solution, 2. bioreactor Applicon (5 L), 3. optical fibres, 4. lens with sensitive layer, 5. light source and the detector (*λ_EX_* = 470 nm, *λ_EM_* = 580 nm), 6. PC LabView, 7. control unit, 8. peristaltic pump, 9. temperature probe of NeoFox.

**Figure 3 sensors-21-00565-f003:**
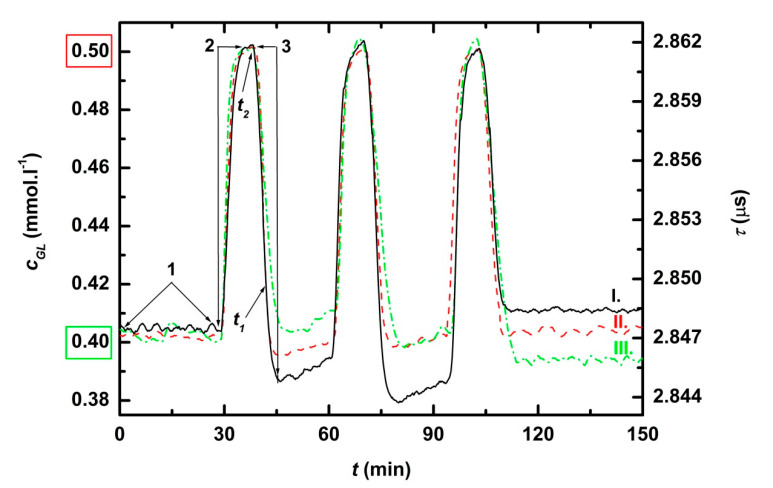
Time record of the feedback regulation of lower glucose concentration (dilution with buffer) for three independent experiments at the same conditions (I.-III.). After the initial three *t_chec_* (position 1), the concentration of glucose was increased from *c_GL_^DES^* = 0.4 mM to *c_GL_^MAX^* = 0.5 mM by hand pipetting of concentrated glucose solution into the bioreactor (position 2). After *c_GL_^MAX^* detection, the pump of feedback loop dosed buffer into the reactor (position 3). *t_1_* is time when actual measured glucose concentration exceeded glucose concentration after regulation for 10% and *t_2_* is time when buffer was added.

**Figure 4 sensors-21-00565-f004:**
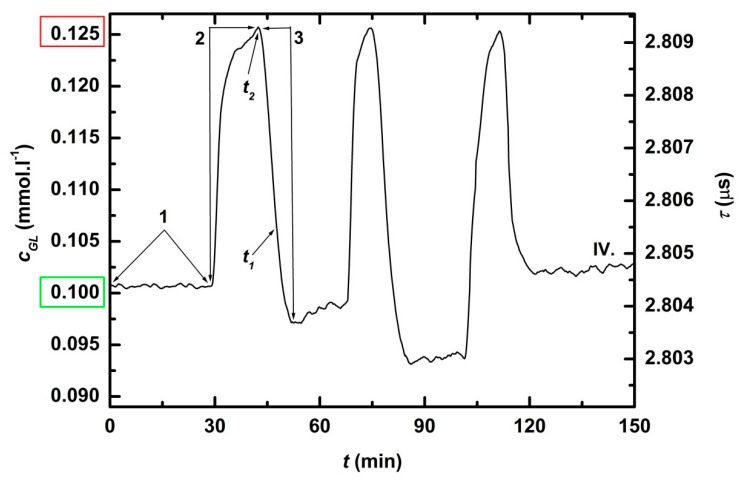
Time record of the feedback regulation of lower glucose concentration (dilution with buffer, test IV). After the initial three *t_chec_* (position 1), the concentration of glucose increased from *c_GL_^DES^* = 0.100 mM to *c_GL_^MAX^* = 0.125 mM by hand pipetting of concentrated glucose solution into the bioreactor (position 2). After *c_GL_^MAX^* detection, the pump of feedback loop dosed buffer into the reactor (position 3). *t_1_* is the time when actual measured glucose concentration exceeded glucose concentration after regulation for 10% and *t_2_* is time when buffer was added.

**Figure 5 sensors-21-00565-f005:**
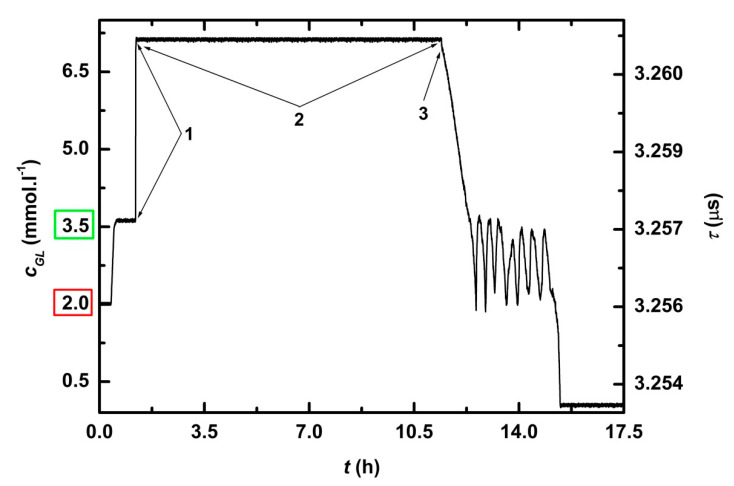
Time record of glucose concentration during fed-batch cultivation of *Saccharomyces cerevisiae*. After two-point calibration (*c_GL_^DES^* = 3.5 mM, *c_GL_^MIN^* = 2 mM), glucose was added to complete YPG medium (*c_GL_* = 111 mM), which was out of the range (0-7 mM) of the biosensor (position 1). The bioreactor was inoculated with an overnight culture of Saccharomyces cerevisiae and the feedback regulation was switched on (position 2). After 11.5 h of fermentation, *c_GL_* dropped below 7 mM (position 3).

**Figure 6 sensors-21-00565-f006:**
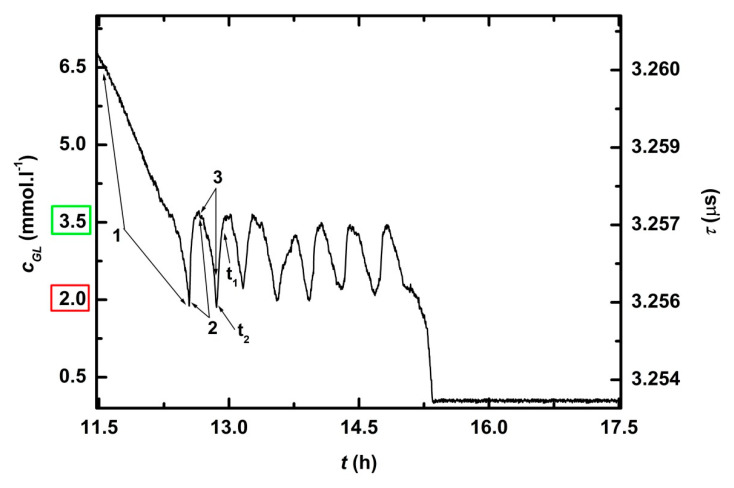
Detailed time record of feedback regulation of glucose concentration during stationary phase of cultivation of *Saccharomyces cerevisiae*.

**Figure 7 sensors-21-00565-f007:**
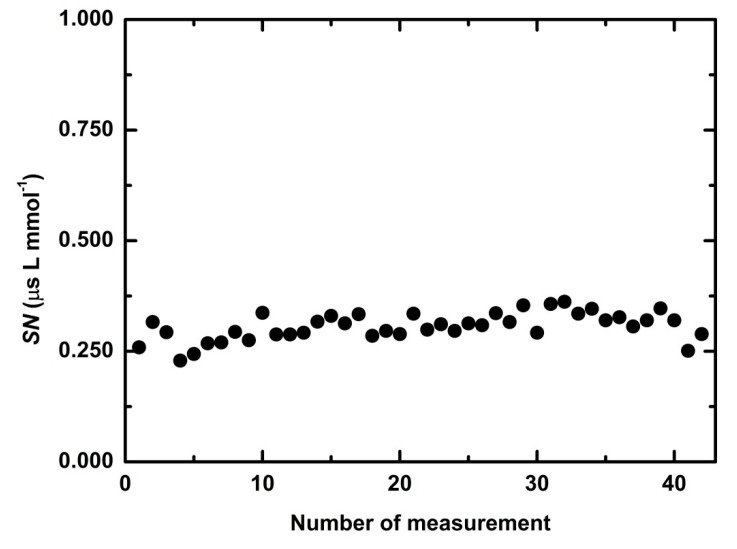
Sensitivity of the optical sensitive layer in 42 repeated measurements during two months.

**Figure 8 sensors-21-00565-f008:**
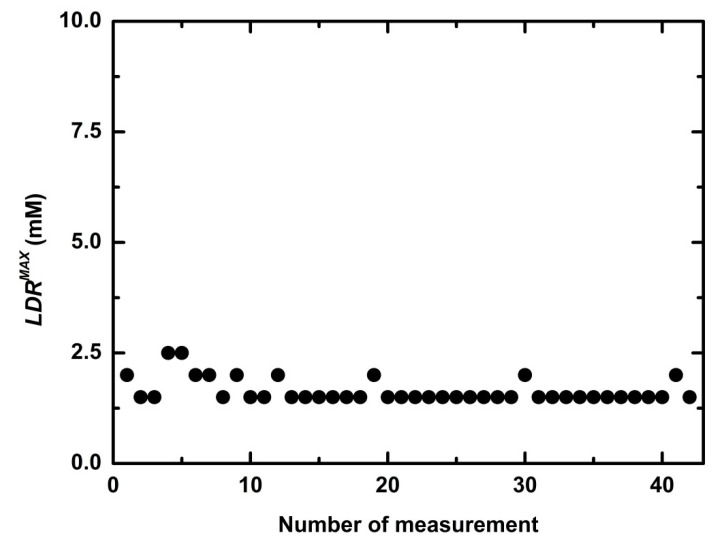
Maxima of linear dynamic range of the optical sensitive layer in 42 repeated measurements during two months.

**Figure 9 sensors-21-00565-f009:**
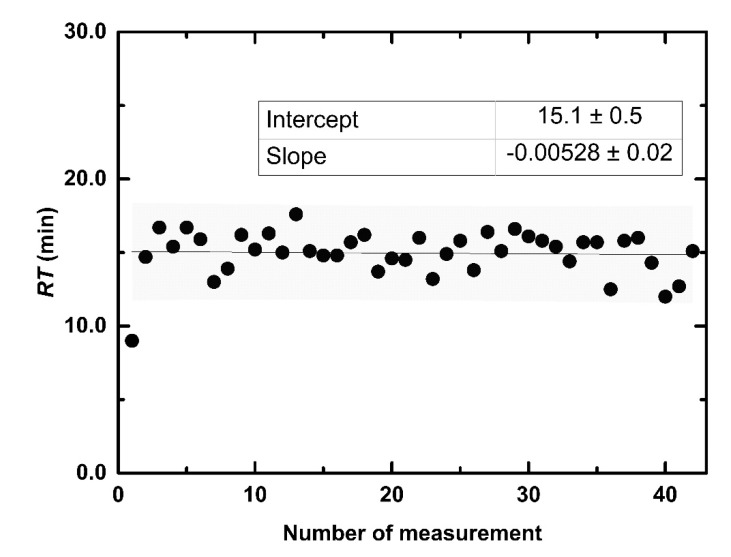
Response time of the optical sensitive layer in 42 repeated measurements during two months.

**Table 1 sensors-21-00565-t001:** Parameters and analytical characteristics of sensitive layers.

Test	Velocity of Stirring in Activation of Sepabeads^®^(rpm)	Enzyme (mg_enzyme_/g_SEPA_)		Thickness * (nm)	*SN* (μs L mmol^−1^)	*LDR* (mM)	*RT* (min)
		GOX^X-S^	GOX^II-S^				
Dilution	20	125	-----	280	0.452	0–1.5	3
Cultivation	20	-----	12.5	225	0.091	0–7	5
Response reproducibility **	800	125	-----	300	0.306	0–1.6	9

* The average of thicknesses measured with the microscope Tescan. ** Average values of *SN*, *LDR*, and *RT* during testing of reproducibility of biosensor response.

**Table 2 sensors-21-00565-t002:** Response times of the biosensor and deviations from desired glucose concentration in case of step increase of glucose concentration.

Experiment	*c_GL_^DES^* (mM)	*c_GL_^REG^* (mM)	*RT_90_* (min)	*s* (%)
I.	0.4	0.388	3	3
		0.380	4	5
		0.411	4	3
II.	0.4	0.404	2	1
		0.400	4	0
		0.393	4	2
III.	0.4	0.396	3	1
		0.398	5	1
		0.400	5	0
IV.	0.1	0.097	5	3
		0.093	5	7
		0.101	2	1

**Table 3 sensors-21-00565-t003:** Response times and deviation from desired glucose concentration during feedback control of glucose concentration in stationary phase of cultivation of *Saccharomyces cerevisae.*

Number of of cycle ↓↑	*c_GL_^DES^* (mM)	*c_GL_^REG^* (mM)	*RT_90_* (min)	*s^*^* (%)
1	3.5	3.7	6	6
2	3.5	3.6	3	3
3	3.5	3.6	3	3
4	3.5	3.2	6	9
5	3.5	3.5	4	0
6	3.5	3.4	2	3
7	3.5	3.4	4	3

Deviation (*s^*^*) defined in Equation (4).

## Data Availability

The data presented in this study are available on request from the corresponding author. The data are not publicly available due to Pat. No. CZ30355 / PUV 2016-33183.

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
