# Peer review of "Cultivation of Saccharomyces cerevisiae with Feedback Regulation of Glucose Concentration Controlled by Optical Fiber Glucose Sensor"

_sensors, 2021, doi:10.3390/s21020565_

Round 1

Reviewer 2 Report

This manuscript details the cultivation of Saccharomyces cerevisiae with feedback regulation of glucose concentration controlled by optical fiber glucose sensor. The paper is well-written, easy to follow and the authors have completed an appropriate set of experiments. However, some issues should be considered in the methodology and discussion of results

  1. What was the concentration of the enzyme solution used in the immobilization process?
  2. Stirring conditions might promote enzyme loss from the polymer network. As part of the characterization of the optically sensitive layers, was any test of enzyme desorption performed? If so, what was the result of this test?
  3. On the other hand, it is known the toxicity that ruthenium complexes possess, what implications would have the release or desorption of this compound in the culture medium? Was any test performed to verify the stability of the compound in the polymer network? Could some other compound be used as an optical transducer?

Round 2
